# Transparent tissue in solid state for solvent-free and antifade 3D imaging

Fu-Ting Hsiao[1,9], Hung-Jen Chien [1,9], Ya-Hsien Chou[1,2], Shih-Jung Peng[2], Mei-Hsin Chung[1,3], Tzu-Hui Huang[1], Li-Wen Lo[4], Chia-Ning Shen [4,5], Hsiu-Pi Chang[6], Chih-Yuan Lee[6], Chien-Chia Chen [6], Yung-Ming Jeng [7], Yu-Wen Tien [6] & Shiue-Cheng Tang [1,2,8] ✉

Optical clearing with high-refractive-index (high-$n$) reagents is essential for 3D tissue imaging. However, the current liquid-based clearing condition and dye environment suffer from solvent evaporation and photobleaching, causing difficulties in maintaining the tissue optical and fluorescent features. Here, using the Gladstone-Dale equation [$(n-1)$/density=constant] as a design concept, we develop a solid (solvent-free) high-$n$ acrylamide-based copolymer to embed mouse and human tissues for clearing and imaging. In the solid state, the fluorescent dye-labeled tissue matrices are filled and packed with the high-$n$ copolymer, minimizing scattering in in-depth imaging and dye fading. This transparent, liquid-free condition provides a friendly tissue and cellular environment to facilitate high/super-resolution 3D imaging, preservation, transfer, and sharing among laboratories to investigate the morphologies of interest in experimental and clinical conditions.

Paraffin embedding is routinely used in medical laboratories to solidify and preserve tissues (e.g., formalin-fixed surgical specimens) for classic microtome-based histology. The solid wax strongly scatters light; thus, the wax-tissue composite is opaque prior to microtome sectioning and dewaxing for staining and microscopy. Alternatively, in modern 3D histology, investigators immerse the whole-mount specimen in the high-refractive-index (high-$n$) solution to reduce scattering for 3D confocal or light-sheet microscopy[1–5]. The refractive index $n$ matching between the solution and the tissue/cellular components (proteins, DNA, and lipids; $n \geq 1.46$)[6] facilitates photon penetration, thereby enabling the fluorescently labeled tissue structures, such as the neurovascular networks, to be visualized in a 3D space continuum[7–10]. However, as with any liquid solution, the solvent of the clearing solution gradually evaporates, leading to solute precipitation and/or changes in chemical and optical conditions. There are also safety concerns when toxic or corrosive solvents (e.g., benzyl alcohol/benzyl benzoate, known as BABB) involved in the clearing and imaging

processes evaporate or leak to the environment. In addition, in 3D microscopy, a liquid holder as well as specimen mounting to the holder is required to maintain the $n$ matching condition (e.g., the sample chamber in light-sheet microscopy). These constraints, compared with the streamlined clinical 2D histology, largely limit the use of 3D microscopy as a standard tool in tissue analysis.

To relieve the constraints imposed by the clearing liquid, preparation of transparent tissues in a solid, liquid-free condition provides an attractive strategy to fundamentally change the current setting for 3D tissue imaging. In the solid state, an extra advantage is that the firm and stable environment hinders chemical reactions (e.g., photobleaching) by limiting the diffusional contact between the indicator dye and reactive species (e.g., oxygen)[11,12], thereby improving the shelf life and photochemical stability of the fluorescently labeled specimens. The photostability is particularly important in high-resolution 3D imaging (e.g., super-resolution microscopy) due to the prolonged laser exposure time under the high-power objective[13].

[1]Institute of Biotechnology, National Tsing Hua University, Hsinchu, Taiwan. [2]Department of Medical Science, National Tsing Hua University, Hsinchu, Taiwan. [3]Department of Pathology, National Taiwan University Hospital-Hsinchu Branch, Hsinchu, Taiwan. [4]Biomedical Translation Research Center, Academia Sinica, Taipei, Taiwan. [5]Genomics Research Center, Academia Sinica, Taipei, Taiwan. [6]Department of Surgery, National Taiwan University Hospital, Taipei, Taiwan. [7]Department of Pathology, National Taiwan University Hospital, Taipei, Taiwan. [8]Department of Chemical Engineering, National Tsing Hua University, Hsinchu, Taiwan. [9]These authors contributed equally: Fu-Ting Hsiao, Hung-Jen Chien. ✉e-mail: sctang@life.nthu.edu.tw

Interestingly, similar to a variety of hydrophobic and hydrophilic high-$n$ clearing liquids[14–16], different classes of high-$n$ polymers (e.g., phosphorus-, sulfur-, and halogen-containing polymers) have been identified and fabricated for optical applications, such as the micro-lenses array in complementary metal oxide semiconductor (or CMOS) sensors[17–19]. In bioimaging applications, however, the solid high-$n$ polymers have not been explored and tested for tissue embedding and polymerization. Particularly, in polymerization, the non-physiological reaction conditions, such as high temperature or hydrophobicity, could denature and/or bleach the fluorophores used in tissue staining, affecting the subsequent morphological characterization.

Among the tissue-friendly polymers, the low-density poly-acrylamide gel has long been used in electrophoresis and recently in the CLARITY/hydrogel approach to support the lipid-extracted tissue in the clearing solution for 3D microscopy[20–22]. By adjusting the hydrogel's composition, the chemical environment of the tissue-hydrogel complex, such as the electrostatic interactions, can be engineered to create a favorable condition for tissue clearing[23,24]. At a higher density, the transparent polyacrylamide and acrylamide-based polymers have also been explored for ophthalmological applications[25–27]. The high-$n$ acrylamide-based polymers, in particular, provide a unique opportunity for simultaneous tissue embedding and clearing at a solid state to replace the high-$n$ liquid in 3D imaging. Importantly, because the synthesis of polyacrylamide can be controlled by ultraviolet radiation, the photo-polymerization can be directly performed on the microscope glass to integrate the processes of tissue embedding, clearing, and 3D imaging with the same polymerized tissue slide.

Here, using an efficient two-step approach, we establish a tissue clearing method via tissue embedding in the high-$n$ copolymer of acrylamide and n-hydroxymethyl acrylamide (or A-ha copolymer) for 3D fluorescence imaging with minimal photobleaching.

## Results

### High-$n$ acrylamide-$co$-n-hydroxymethyl acrylamide copolymer

The refractive index $n$ can be considered as the factor by which the speed of light is reduced with respect to its vacuum value. The empirical Gladstone-Dale equation[28], $(n-1)/\rho$ = constant, further suggests that the increase in refractive index of a material (or reduction in light speed in material) vs. vacuum ($n = 1$) is proportional to the density $\rho$. In polymer science, the macromolecules formed via covalent bonding are generally denser (higher $\rho$, and thus higher $n$) than the monomers in solution prior to polymerization. The bonding reaction (polymerization) thus provides an effective way to increase the density of an organic mixture, creating a high-$n$ environment for optical imaging.

Using the Gladstone-Dale equation as a design concept, we synthesized the high-$n$ poly(acrylamide-$co$-n-hydroxymethyl acrylamide), or A-ha copolymer, by first linearly increasing the $n$ to $1.475 \pm 0.001$ (twelve repeats) with incremental increase in the monomers in water (Fig. 1a), and then a jump of $n$ to $1.533 \pm 0.005$ (twenty-four repeats) via photo-polymerization at 86.7% monomer mass fraction. This ultrahigh monomer-to-water mass ratio (6.5–1, Supplementary Movie 1) was achieved by using a ternary solution of acrylamide (35.8%) and n-hydroxymethyl acrylamide (50.9%) in water (molar ratio 1:1), which avoids precipitation encountered in a binary acrylamide-water solution at the same mass fraction. Once polymerized, the A-ha copolymer is rigid and has a density at $1.30 \pm 0.01$ (g/cm$^3$, 20 repeats), an 18.1% increase compared with that of the monomer solution.

In polymerization, because cross-linkers (e.g., bisacrylamide) were not added as a reactant, the A-ha copolymer is thermoplastic and requires monomers at ≥76.0% in mass fraction to form a rigid material (Fig. 1b and Supplementary Fig. 1, vs. soft polyacrylamide gel). Importantly, the rigid A-ha copolymer can be fabricated into different shapes for tissue clearing (i.e., increase in transmittance) and imaging

applications (Fig. 1c–k). To evaluate the optical property, Fig. 2 and Supplementary Fig. 2 (extended list) compare the refractive indexes of the high-$n$ polymers (A-ha, sulfur-, phosphorus-, and halogen-containing polymers) with those of the cellular components (proteins, DNA, and lipids) and published clearing reagents. Interestingly, because proteins and DNA can also be considered as copolymers with sulfur or phosphorus atoms, their estimated refractive indexes (≥1.54)[29–31] are closer to the high-$n$ polymers than to the majority of the clearing reagents.

### Two-step tissue clearing and 3D imaging with A-ha copolymer

To prepare transparent tissues, we established a two-step approach to increase the transmittance of the specimen of interest (Fig. 3a–g). First, we immersed the tissue (500-µm mouse kidney section) in the A-ha monomer solution ($n = 1.48$) for an hour to elevate the transmittance to $64.4\% \pm 9.0\%$ (wavelength 640 nm) from $4.9\% \pm 4.4\%$ in PBS ($n = 1.33$). This was followed by photo-polymerization to synthesize the A-ha copolymer and transformed the immersed tissue to become an embedded specimen. The new optical condition ($n = 1.53$) further increases the transmittance to $93.5\% \pm 5.0\%$ (wavelength 640 nm) of the same specimen. Importantly, the second step of the clearing process is efficiently achieved within 30 minutes because the free radical photo-polymerization has fast kinetics. In the process, we observed that the size of kidney increases from 100% (PBS) to $115 \pm 1\%$ (twelve repeats in A-ha monomer solution, 1st step) to $118 \pm 2\%$ (twelve repeats in A-ha copolymer, 2nd step), indicating a mild tissue expansion caused by the immersion and embedding processes (Supplementary Fig. 3).

The compatibility of A-ha copolymer with fluorescence imaging is shown in Fig. 4a–j. At the organ level, the A-ha-cleared mouse liver with nuclear and neurovascular labeling is prominently seen under the confocal microscope. In particular, the panoramic 2D image (Fig. 4c–e) and 3D projection (Fig. 4f, g) detect and identify the hepatic nerve plexus following the portal vein and its bifurcation in extension. Furthermore, using the high-power objective, we specified the detailed hepatic nervous and endothelial structures. Figure 4h–j and Supplementary Movie 2 (in-depth recording) present the perivascular sympathetic innervation with subcellular-level resolution, confirming the compatibility of A-ha copolymer with high-resolution microscopy. Additional examples of 3D fluorescence imaging with A-ha copolymer are presented in Supplementary Fig. 4 and Supplementary Movie 3 and 4 (mouse brain and kidney).

### Multimodal cm-to-subcellular human pancreas imaging with A-ha

In clinical tissue analysis, we used the human donor pancreas with low-grade duct lesions (pancreatic intraepithelial neoplasia or PanIN)[32,33] to demonstrate the integration of A-ha-based 3D histology with the clinical 2D histology. In Fig. 5a–j, a side-by-side display of the adjacent vibratome and microtome sections (the former in A-ha, the latter in paraffin before sectioning) is used to illustrate the matched pancreatic lesion environment in the stereomicroscopic, hematoxylin and eosin (H&E), and fluorescence images. Specifically, in the A-ha copolymer, the transparent vibratome section enables in-depth neurovascular imaging of human pancreas with paired S100B/glial and CD31/endo-thelial fluorescent labeling (Fig. 5e–j), whereas in the microtome section, the gold-standard H&E stain specifies the duct lesion (Fig. 5d).

Clinically, using the 2D H&E histology, prior research has established that the PanIN lesion is associated with parenchymal (acinar) atrophy and fibrosis[34,35]. Here, using the 3D images (Supplementary Movie 5), we detect that the PanIN and peri-PanIN stroma are also associated with S100B$^+$ glial/nerve bundles (Fig. 5e–g, 5- to 50-µm-thick; Supplementary Movie 6 shows peri-lesional ganglion with neurons enclosed by glia)[8,36]. In the same region, the atrophic parenchyma is linked to less prominent CD31$^+$ endothelium vs. adjacent normal

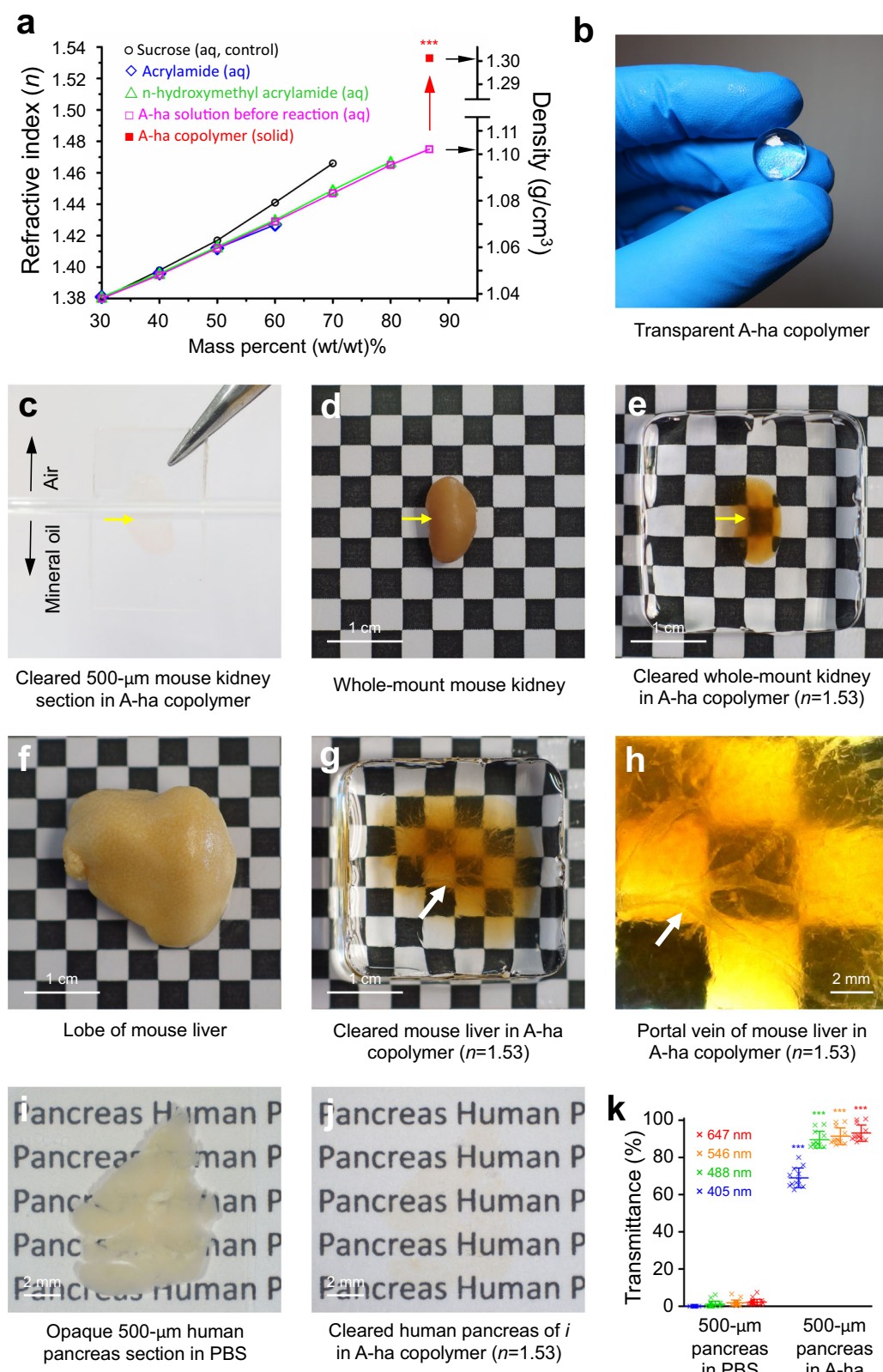

**Fig. 1 | Synthesis of high-*n* A-ha copolymer for tissue clearing. a** Increases in refractive index *n* and density at high mass fraction of monomers (acrylamide and n-hydroxymethyl acrylamide) and after photo-polymerization (arrow). ≥12 repeats per data point. ***$p < 0.001$ vs. A-ha monomer solution (two-sided unpaired Student's *t* test). **b** Rigid and transparent high-*n* A-ha copolymer. **c**–**k** Optically cleared tissues in A-ha copolymer (representative images). **c**–**e** mouse kidney (yellow arrow). **f**–**h** mouse liver (enlarged in *h*; arrow, portal vein). **i, j** vibratome section of human pancreas in PBS vs. A-ha copolymer. The drastic increase in transmittance in A-ha is quantified in **k** (n = 21 and 12 independent measurements in PBS and A-ha conditions, respectively; ***$p < 0.001$ vs. PBS, two-sided unpaired Student's *t* test). Data are presented as means ± standard deviation. Source data are provided as a Source Data file.

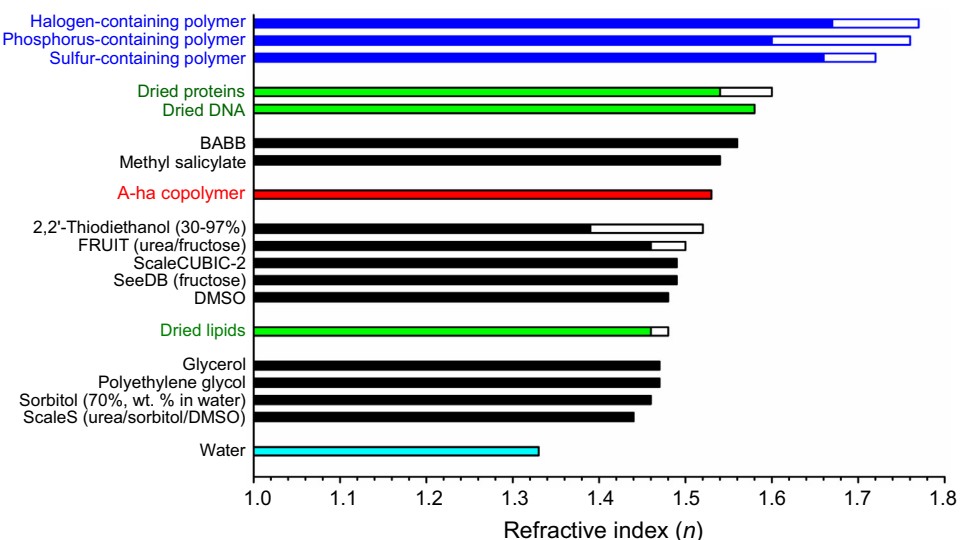

**Fig. 2 | Comparison of refractive indexes among high-*n* polymers, cellular components, and published tissue-clearing reagents.** Empty bar indicates the range of refractive indexes of a polymer, cellular component (protein, DNA, and lipid), or reagent. An extended list of the clearing reagents is provided in Supplementary Fig. 2 for detailed comparison. Source data are provided as a Source Data file.

lobules (Fig. 5h–j). These two features–preserved and revealed in the A-ha copolymer–appear to be the signs of peri-lesional neural association and reduced vascular supply to the atrophic lobule. Comparable but drastic changes of pancreatic lobules, including loss of acini, fibrosis, neural invasion, and lack of vascular supply, have been previously reported in pancreatic cancer[37–40].

In addition to the change in the neurovascular association, we also identified the endocrine pancreas remodeling around the exocrine duct lesion. This phenomenon is highlighted by the detection of duct-islet cell clusters (Fig. 5k–o and Supplementary Movie 7)[33,41]. In the A-ha copolymer, the cluster is well preserved and can be identified with glucagon (α-cell), insulin (β-cell), and cytokeratin 7 (CK7, duct cell) staining. When enlarged, the endocrine α- and β-cells were seen juxtaposed with the exocrine duct cells (Fig. 5l), a sharp contrast to the discrete units of islets and ducts in the normal lobules. Furthermore, we used the Airyscan detector array microscopy (Carl Zeiss, super-resolution mode)[42] to enhance the resolving power of cellular imaging in the A-ha copolymer. In the duct-islet cell cluster, the glucagon⁺ and insulin⁺ vesicles were identified in the α- and β-cells, respectively, adjacent to the CK7⁺ duct cells (Fig. 5m–o). Technically, this series of human pancreas imaging–from duct lesion to islet cells to vesicles–demonstrates the multimodal, multidimensional, and multiscale approaches of imaging with the A-ha copolymer.

### Antifade and super-resolution 3D human pancreas imaging

One fundamental difference of fluorescence imaging in the high-*n* A-ha copolymer vs. high-*n* clearing solution is photochemical stability. This distinct property can be visualized by time-series confocal imaging of the fluorescence derived from direct immunohistochemistry. We used the human pancreas vasculature (due to its abundance, e.g., Fig. 5h) labeled with the sensitive Alexa Fluor (AF) −647-conjugated anti-CD31 primary antibody to demonstrate this phenomenon.

In Fig. 6, we compare the decreases in AF-647 signals in time-series confocal imaging while the specimens were embedded in the A-ha copolymer, immersed in the aqueous Ce3D™ solution ($n = 1.5$, Biolegend; tissue clearing control)[43], or immersed in the organic solvent-based CytoVista™ clearing solution ($n = 1.5$, ThermoFisher; tissue clearing control)[44,45]. The less sensitive indirect immunohistochemistry (anti-S100B labeled with multiple AF-546-secondary antibodies, e.g., Fig. 5e) and chemical stain of nuclei (DAPI) were used as the positive controls to monitor the fluorescence signals. Importantly, after 500

frames were taken from the solid and liquid clearing conditions (90-min. continuous imaging of a 320×320-μm region), the mean AF-647 fluorescence in A-ha decreased by $9 ± 2\%$ (Fig. 6a–c), which was a marked difference compared with the $95 ± 4\%$ and $55 ± 7\%$ decreases of the same indicator dye in the Ce3D™ and CytoVista™, respectively (Fig. 6d–f, g–i; six repeats; $p < 0.001$). The antifade feature of fluorescence imaging with A-ha copolymer is presented in Supplementary Movie 8. The comparison of AF-647 stability when embedded in A-ha vs. immersed in liquids applied in the iDISCO (dibenzyl ether)[46], CLARITY (FocusClear)[20], and CUBIC (sucrose/urea/triethanolamine solution)[3] -based clearing methods is presented in Supplementary Fig. 5a–l. Supplementary Movie 9 shows the side-by-side and frame-by-frame comparison of the four conditions. The intrinsic stability of nuclear dyes–DAPI (positive control), Hoechst, and propidium iodide–in A-ha and Ce3D is presented in Supplementary Fig. 5m, n.

Because the A-ha copolymer enables both antifade and 3D fluorescence imaging, we next applied multimodal imaging (as illustrated in Fig. 5) with in-depth Airyscan imaging (super-resolution mode) to resolve the unique 3D neurovascular features of human pancreas, which otherwise cannot be effectively achieved with the clinical 2D histology or 3D imaging in the clearing liquid. Figure 7 is an example of this approach, featuring the human pancreas lobule-to-nucleolus magnification, starting from the gross view of vascular pathway (Fig. 7a, b; CD31⁺) and ending at the dimly stained neuronal nucleus in the intrapancreatic ganglion (Fig. 7c–g; also see Supplementary Movie 10–12). Overall, the antifade and solvent-free conditions of A-ha create a 3D imaging platform that mimics the chemical (stable) and optical (transparent) environments of clinical 2D histology for high/super-resolution 3D imaging on the standard microscope slide (Supplementary Fig. 6).

### Discussion

High-resolution 3D fluorescence imaging prefers stable chemical and optical conditions to resolve the dye-labeled tissue and cellular structures in health and disease. Although major advances in the machine (e.g., detector and illumination strategy), indicator dye (e.g., quantum dot), and suppression of scattering (tissue clearing) have improved the resolving power in in-depth imaging, the intrinsic volatility and molecular diffusion of liquids limit the current approach and design for 3D histology. Here, using the high-*n* A-ha copolymer, we created a transparent environment for 3D tissue imaging in a solvent-

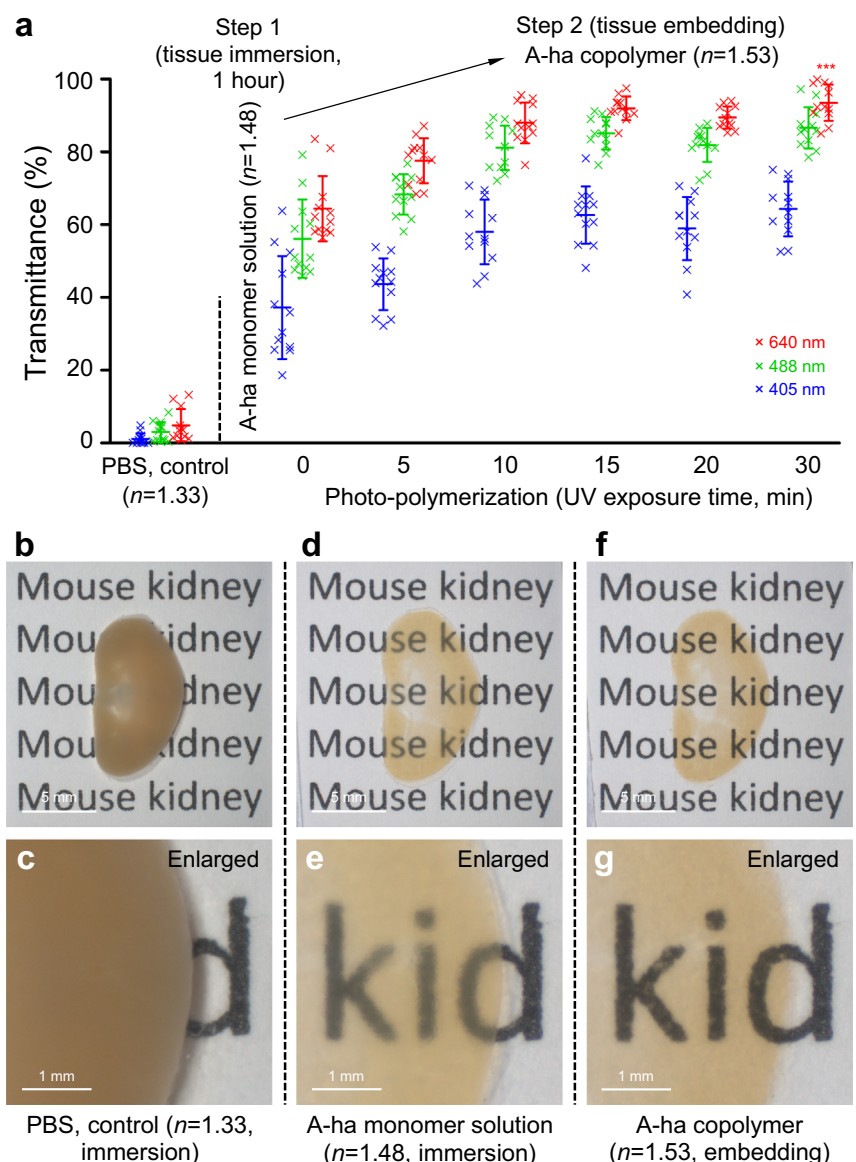

**Fig. 3 | Two-step tissue clearing with A-ha copolymer. a–g** Mechanism of A-ha-based tissue clearing. Test tissue: 500-μm-thick mouse kidney. Control: tissue in PBS. Step 1 shows a first increase in transmittance after tissue immersion in the A-ha (acrylamide and n-hydroxymethyl acrylamide) monomer solution. A second increase occurs after polymerization (step 2, embedding; achieved within 30 min). Twelve repeats for each condition. ***$p < 0.001$ vs. time = 0 (two-sided unpaired Student's *t* test). Data are presented as means ± standard deviation. **b–g** are representative images of the two-step clearing process. **c**, **e**, **g** are enlarged images of **b**, **d**, **f** (center), respectively. The increase in transmittance reveals the word at the back of the specimen. Source data are provided as a Source Data file.

free condition. The tissue embedding was made possible by photo-polymerization at room temperature in an acrylamide-based solution, minimizing concerns of tissue damage and artifacts created in a harsh condition of polymer synthesis.

Importantly, using the experimental and clinical specimens, we established the organ-to-nucleus and panoramic-to-super-resolution imaging, respectively, confirming the intact macro- and micro-structures of the embedded tissues in the A-ha copolymer. Particularly, using the human pancreas, we demonstrated the matched duct lesion microenvironment in the stereomicroscopic, fluorescence, and gold-standard H&E images, highlighting the 3D/2D multimodal imaging with the A-ha copolymer in a clinically related setting. It should be noted that because the standard organic dyes (DAPI and Alexa Fluor) are stable in the solid A-ha copolymer, but not the fluorescent proteins, which evolve in an aqueous environment, our work is designed for 3D histology following standard (direct or indirect) immunohistochemistry. In immunohistochemistry, a stable dye environment benefits the

use of high/super-resolution imaging by minimizing photobleaching in procedures that require concentrated laser excitation and/or pro-longed imaging time to record in-depth 3D signals (e.g., Fig. 7 and Supplementary Movie 10–12).

In 3D signal detection, because false-negative results are fre-quently encountered in fluorescent immunohistochemistry, the anti-fade vascular (anti-CD31) and nuclear (DAPI) labeling developed in this research provides the important positive controls for clinical tissue examination–due to their abundance–to monitor the penetration of antibody and indicator dye in 3D labeling. In the imaging facility core, the antifade controls also offer a suitable chemical environment to compare different super-resolution methods (e.g., Airyscan vs. stimu-lated emission depletion, or STED, microscopy) via a standard 3D specimen (e.g., Supplementary Fig. 6c).

From the logistic perspective, because the indicator dyes are stable in the solid A-ha copolymer, the improved shelf life allows transfer of 3D specimens among laboratories for off-site imaging (e.g.,

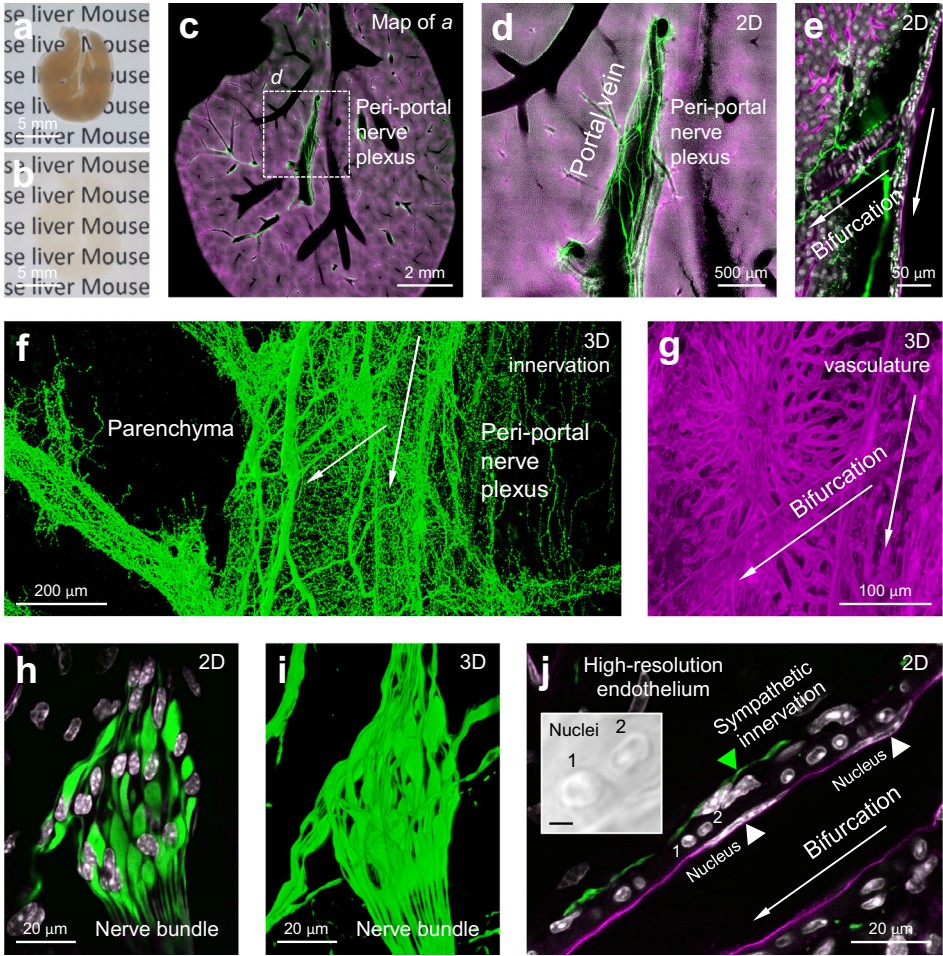

**Fig. 4 | Mouse liver organ-to-nucleus imaging with A-ha copolymer. a, b** A-ha-based tissue clearing of 500-μm-thick mouse liver (representative images of three livers). The increase in transmittance enables in-depth fluorescence imaging of sympathetic nerves (anti-tyrosine hydroxylase stain, green), blood vessels (perfusion labeling, magenta), and nuclei (DAPI stain, white) shown in **c–j**. **c–j** Representative 2D images and 3D projections of mouse liver neurovascular network. Arrows in **e–g** and **j** indicate the same bifurcation of portal vein. **h–j** show the subcellular features of nerve bundle and endothelium preserved in the A-ha copolymer. In panel **j**, green and white arrowheads indicate the perivascular sympathetic nerve fiber and elongated endothelial nuclei, respectively. Transmitted light signals of nuclei (#1 and #2) embedded in A-ha are magnified in inset (bar, 2 μm).

at microscope center) and/or tissue preservation and archiving, alleviating the constraints of facility and time in 3D tissue analysis. In particular, sharing of the antifade 3D tissues benefits resource and information exchanges in the pandemic/post-pandemic era, in which international interactions and in-person visits are significantly reduced. In terms of safety in tissue clearing, the replacement of organic solvents with solid polymers in 3D imaging protects staffs as well as machines in the microscope room, in which solvent evaporation or leakage is a potential concern compared with the standard 2D imaging. Regarding safety in polymer synthesis, we would like to stress that the monomers of A-ha copolymer–acrylamide and n-hydroxymethyl acrylamide–are toxins when in solution (especially acrylamide), so care must be taken to avoid direct contact with the monomer solution. Once polymerized, similar to the polyacrylamide gel used in electrophoresis, the A-ha copolymer is chemically stable (long shelf life) with strong mechanical strength (provided by ultrahigh monomer content, Fig. 1a) to protect the cleared 3D specimen in transport and imaging.

In conclusion, before this research, 3D tissue imaging has been limited by solvent and dye instability in a liquid environment. Using the Gladstone-Dale equation as a design concept, we develop a solid (solvent-free) high-*n* A-ha copolymer to embed mouse and human tissues for clearing and antifade imaging. This solid microscopic environment offers a friendly and versatile platform that integrates with standard fluorescent dye labeling, clinical 2D histology, and 3D super-resolution imaging for investigation of the cellular structures and neurovascular networks in health and disease.

## Methods
### Ethical statement
Collection and use of human tissues were approved by the Institutional Review Board of National Taiwan University Hospital (NTUH-REC No.: 202107126RINC). Human pancreases were obtained from deceased organ donors whose pancreases were declined for transplantation but whose next of kin had given written consent for use in research. The Institutional Animal Care and Use Committees at National Tsing Hua University approved all procedures with mice (Approval No.: 110037).

### Synthesis of high-*n* A-ha copolymer
Monomers acrylamide and n-hydroxymethyl acrylamide were purchased from Sigma-Aldrich (St. Louis, MO, USA; A8887) and TCI (Tokyo Chemical Industry, Tokyo, Japan; M0574), respectively (caution: both monomers must be handled in a chemical fume hood). Irgacure 2959 (Sigma-Aldrich, 410896) was used as the photoinitiator for the UV-induced photo-polymerization (Supplementary Fig. 6). A-ha monomer solution was prepared on a pre-warmed plate (60–65 °C) by

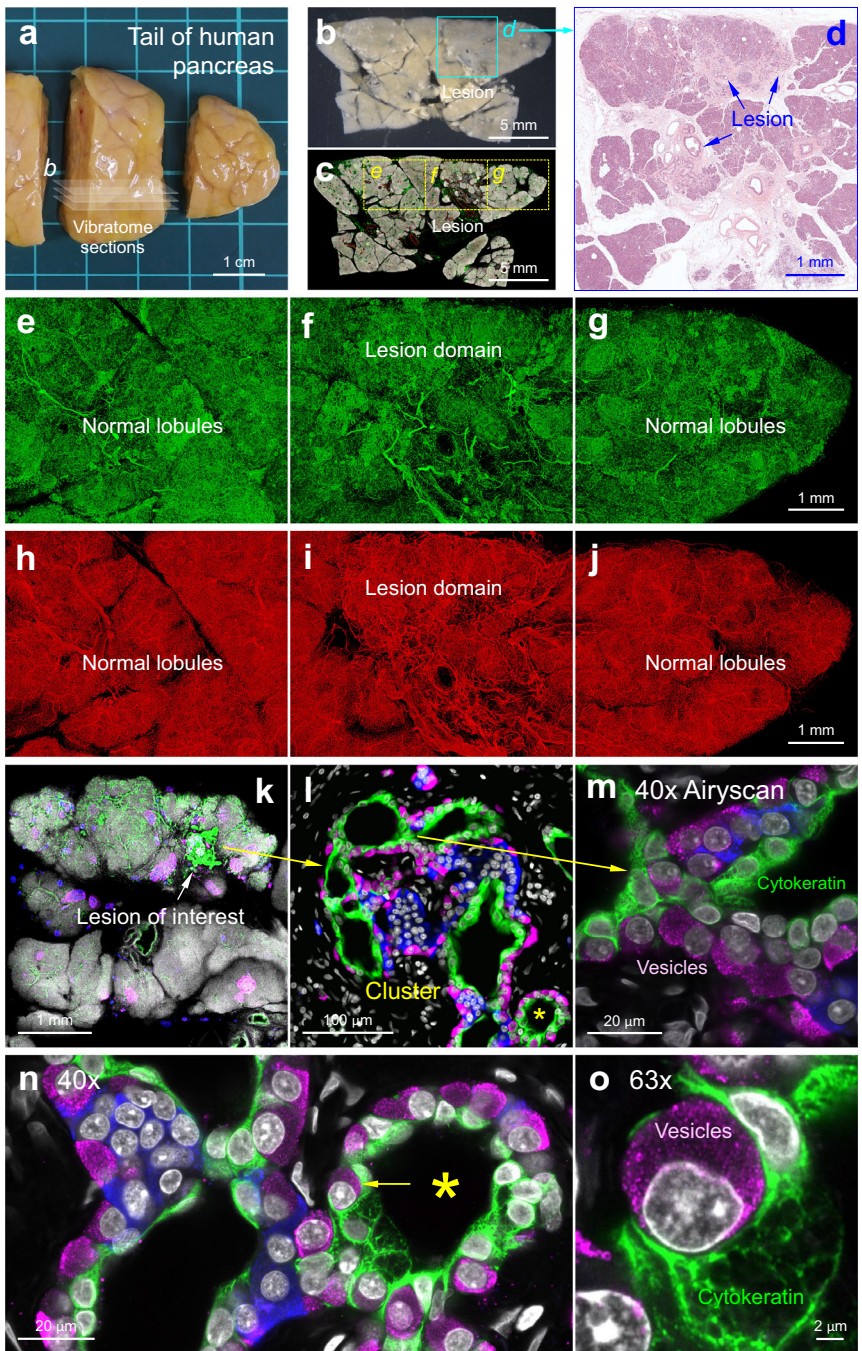

**Fig. 5 | Multimodal cm-to-subcellular human pancreas imaging with A-ha copolymer. a–d** Gross view of human pancreas (tail region, **a**). The pancreas vibratome sections (**a**, **b**) were detected with duct lesions (box in **b**) and confirmed with H&E image (**d**). The vibratome section *b* was labeled with DAPI (nuclei, white), S100B (glia, green), and CD31 (blood vessels, red) and embedded in A-ha for confocal imaging to reveal the microstructure and neurovascular networks (**c**, boxes enlarged in **e**–**j**). **e**–**j** Neurovascular networks of human pancreas (duct lesion vs. normal lobule). Tile scan of **c** allows the use of side-by-side display to reveal the unique neurovascular environment in the lesion domain. Note that adipocytes are also S100B positive[49]. Projection depth, 350 μm. **k**–**o** Peri-lesional endocrine pancreas remodeling preserved and revealed in A-ha. Panel **k** (vibratome section adjacent to **d**) shows the duct lesion. The lesion and the duct-islet cell cluster are enlarged in **l** and **m** (arrows). White, DAPI, nuclei; blue, insulin, β-cells; magenta, glucagon, α-cells; green, cytokeratin 7 (CK7), duct cells. Cytokeratin filaments and vesicles are revealed by Airyscan super-resolution imaging with ×40 objective (**m**). The asterisk in **l** is enlarged in **n** and further magnified in **o** (x63 objective) to confirm the glucagon⁺ vesicles of α-cell (magenta) and cytokeratin filaments of duct cell (CK7⁺, green). **b**–**j** and **k**–**o** are derived from three sets of consecutive vibratome sections. The panels from **a** to **o** (same lesion environment) illustrate the multimodal, multidimensional, and multiscale approaches of human pancreas imaging with A-ha copolymer.

adding 5 g of double-distilled water with 0.3% Irgacure 2959 (0.015 g) to a mixture of 13.48 g of acrylamide and 19.1 g of n-hydroxymethyl acrylamide (Supplementary Movie 1). The solution was stable at room temperature and was sealed in dark prior to reaction. Before and after UV irradiation (30 min), the refractive indexes of A-ha monomer solution and copolymer were measured by a liquid refractometer (Rocker Scientific, Kaohsiung, Taiwan; model: BR92D/BR95D; sucrose, acrylamide, and n-hydroxymethyl acrylamide solutions were used as controls) and a gem refractometer (Fable Gem Refractometer FGR-002A, Guangdong, China), respectively. The photo-polymerization

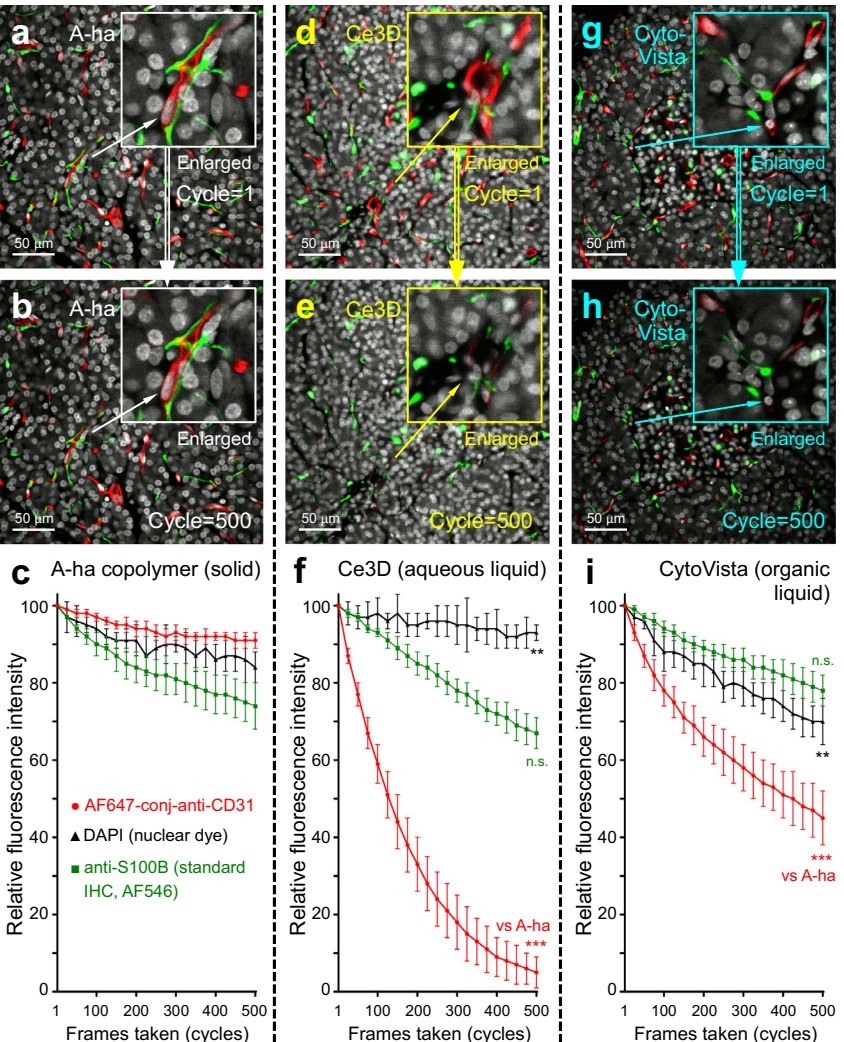

**Fig. 6 | Antifade test of human pancreas imaging with A-ha copolymer.** Time-series confocal imaging of human pancreas in A-ha copolymer (**a**−**c**, solid state), Ce3D clearing liquid (**d**−**f**, aqueous solution), and CytoVista clearing liquid (**g**−**i**, organic solvent). **a**−**c**, **d**−**f**, **g**−**i** are representative images and quantitative analysis of photobleaching. Tissues were labeled with DAPI (white, control), anti-S100B (green, standard immunohistochemistry; control), and anti-CD31 (red, AF-647-conjugated primary antibody; indicator) for the antifade test. ×40 objective was used to acquire 500 frames from a 320 × 320-μm region (-30 μm under tissue surface, six repeats per data point). In **c**, **f**, **i**, data are expressed as % of fluorescence intensity at cycle = 1 (means with standard deviation). ***$p < 0.001$; **$p < 0.01$; n.s., nonsignificant vs. A-ha at cycle = 500 (two-sided unpaired Student's $t$ test). Source data are provided as a Source Data file.

was confirmed by the weak/absent peak of alkenes at 1683 cm$^{-1}$ of A-ha copolymer's FT-IR spectrum (Nicolet iS50, ThermoFisher, Waltham, MA, USA) vs. the spectra of the two monomers (Supplementary Fig. 6d).

## Animal and human tissues

Kidneys and livers harvested from 8 to 12-week wild-type male C57BL/6 mice (National Laboratory Animal Center, Taipei, Taiwan) were used to develop 3D histology with A-ha copolymer. Mice were housed in a controlled environment at 22 °C, 50% humidity, and 12-h light-dark cycles. Human pancreases obtained from deceased organ donors were used to evaluate the compatibility of A-ha copolymer with clinical tissue embedding and imaging. Perfused pancreases from two donors (male, no history of diabetes or pancreatic diseases, aged 40-55, with BMI 20.4-23.6) with normal HbA$_{1c}$, amylase, and lipase levels were used to characterize the pancreatic lobular environment in health and pancreatic intraepithelial neoplasia (PanIN). The two donor pancreases were grossly normal, but low-grade PanIN lesions were detected by organ-wide examination of vibratome sections via

stereomicroscopy; the PanIN lesions were subsequently confirmed by H&E histology[41].

## Tissue preparation and labeling

Blood vessels of mouse liver, brain, and kidney were labeled with cardiac perfusion of lectin-Alexa Fluor 488 conjugates (ThermoFisher) followed by 4% paraformaldehyde perfusion fixation[47,48]. Afterward, the organs were harvested and post-fixed in 4% paraformaldehyde solution for 30 minutes at 15 °C. The human donor pancreas was first cut to separate the head, body, and tail of the organ and then into strips (-1.5 cm in width)[41]. Afterward, the tissue was fixed in formaldehyde 10% (w/v; Macron Fine Chemicals, Centre Valley, PA, USA, H121-08) for two days and then washed in PBS for four days at 4 °C. The fixed mouse and human specimens were later sectioned to 500 μm (or 350 μm in Fig. 5) in thickness by vibratome and transferred to 0.1% paraformaldehyde and 0.25% formaldehyde, respectively, for preservation at 4 °C.

Prior to immunolabeling, the samples were dehydrated by rinsing in methanol/water series (25%−50%−75%−100%) and then immersed in 100% methanol for 5 h. Afterward, the specimens were bleached with

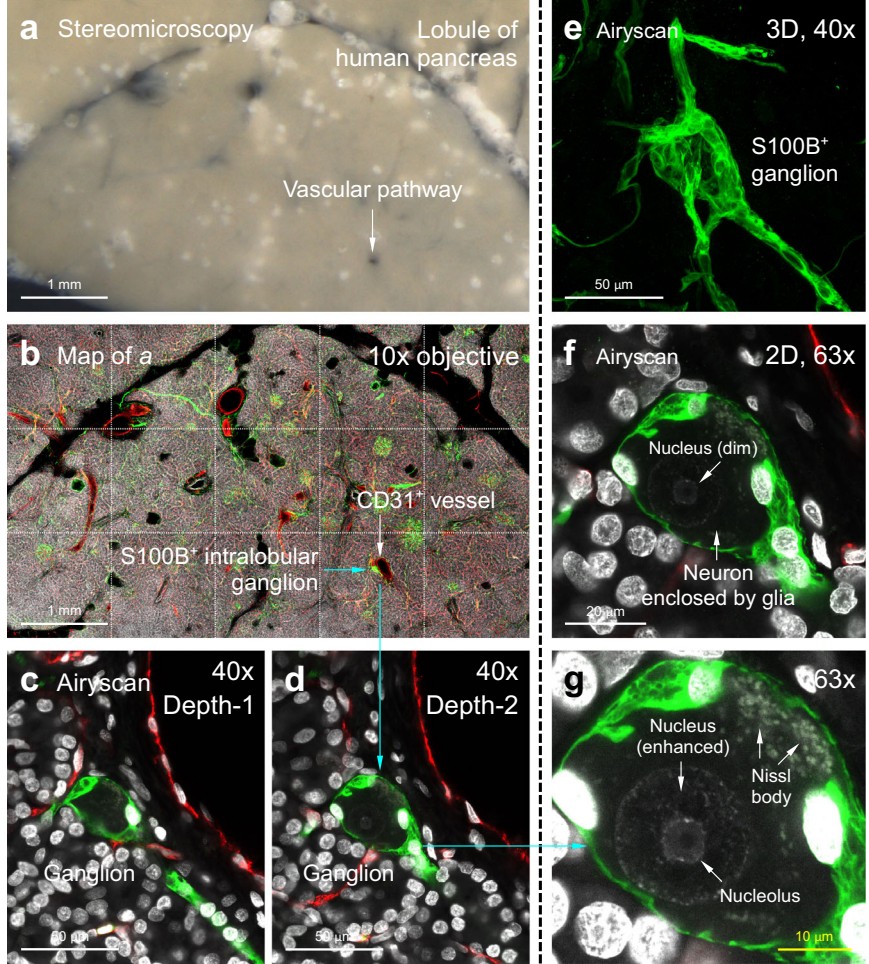

**Fig. 7 | Airyscan 3D super-resolution imaging of human pancreas (representative images of twelve human pancreas vibratome sections). a**, **b** Panoramic view of human pancreatic lobule. Panel **a** (vibratome section) and **b** (confocal image) are derived from the same lobule. Arrows indicate that a ganglion (S100B[+], green) around a vascular pathway (CD31[+], red) is preserved in the A-ha copolymer. White, DAPI, nuclei. **c**–**g** Ganglion examination via in-depth Airyscan to illustrate the glial-neuronal association. **c**–**e** are 2D image and 3D projection of the ganglion (×40 objective). Airyscan with ×63 objective (**f**, **g**) detects and confirms the dimly stained neuronal nucleus (nucleolus at the center). Cyan arrows in **b**, **d**, **g** indicate the lobule-to-nucleolus magnification of human pancreas. Also see Supplementary Movie 10–12 for in-depth Airyscan of the glial-neuronal association.

5% hydrogen peroxide (Sigma-Aldrich, 31642) in 100% methanol overnight at 4 °C. The rehydration process was performed by downgrading serials of methanol/water (25%–50%–75%–100%-PBS). The rehydrated specimens were then immersed in 2% Triton X-100 solution at 15 °C overnight for permeabilization.

Seven different primary antibodies were used alone or in combination to immunolabel the tissues following the protocol outlined below. The antibodies used were rabbit anti-tyrosine hydroxylase (TH, sympathetic marker; AB152, Sigma-Aldrich), rabbit anti-S100B (glial marker; ab52642, Abcam, Cambridge, MA, USA), AF-647-conjugated mouse anti-CD31 (endothelial marker, ab215912, Abcam), mouse anti-glucagon (islet α-cell marker; ab10988, Abcam), AF-488-conjugated mouse anti-insulin (islet β-cell marker; sc-8033_AF488, Santa Cruz Biotechnology, Dallas, TX, USA), rabbit anti-CK7 (epithelial marker; ab68459, Abcam), and AF-647-conjugated rabbit anti-PGP9.5 (neuronal marker; ab196173, Abcam) antibodies.

Before applying the antibodies, the permeabilized tissue sections were rinsed in PBS. This was followed by a blocking step, incubating the tissue with the blocking buffer (2% Triton X-100, 10% normal goat serum, and 0.02% sodium azide in PBS) for 2 h at 15 °C. The primary antibody was then diluted in the dilution buffer (1:100, 0.25% Triton X-100, 1% normal goat serum, and 0.02% sodium azide in PBS) to replace the blocking buffer and incubated for two days at 15 °C. Negative staining controls were prepared by omitting the primary antibody in the buffer.

Alexa Fluor 488, 546, and 647-conjugated secondary antibodies (raised in goat; 1:200, ThermoFisher) were used in combination to reveal the immunostained structures. DAPI (Sigma-Aldrich, D9542) staining was performed to reveal the nuclei. The fluorescently labeled mouse and human specimens were immersed in 4% paraformaldehyde and 10% formaldehyde, respectively, at 4 °C overnight for post-fixation prior to tissue clearing and imaging. Supplementary Table 1 summarizes the immunostaining reagents/dilutions used in the illustrations.

### A-ha-based tissue clearing and 3D microscopy

The vibratome section of mouse or human tissue was first immersed in one well of a 6-well plate with 3 ml of the A-ha monomer solution and placed on a platform rotator for 15 minutes and subsequently transferred to a new well with 3-ml A-ha monomer solution for another 45 minutes. Afterward, a 500-μm-iSpacer (SunJin Lab, Hsinchu, Taiwan) was placed on a coverslip, which was followed by adding fresh A-ha monomer solution (~360 μl) to the spacer compartment and then transferring the tissue from the six-well plate to the spacer. After adding another coverslip on top, the sample was ready for photo-polymerization (Supplementary Fig. 6b). In the clearing kinetics test

(Fig. 3a–g), the 500-μm mouse kidney sections (without labeling) were used to measure the tissue transmittance (Rainbow Light Technology, Transmittance Measuring Equipment TSM-01, Taoyuan, Taiwan) vs. time of UV irradiation (lamp: Philips, TUV PL-L 18W/4P 1CT/25, emission peak at 253.7 nm, irradiance: 9.6 mW/cm$^2$, measured by Accumulated UV Meter UIT-250, Ushio America, Cypress, CA, USA; Supplementary Fig. 6a, b). Thirty minutes of UV irradiation were used as the standard (unless otherwise indicated) to prepare transparent mouse and human specimens for 3D confocal microscopy (Zeiss LSM 800, Carl Zeiss, Jena, Germany; note: A-ha copolymer absorbs water; avoid direct water-A-ha copolymer contact in microscopy) and super-resolution imaging (Zeiss Airyscan, a 32-detector array). The Avizo 6.2 image reconstruction software (VSG, Burlington, MA, USA), Zen software (blue and black editions, Carl Zeiss), and LSM 510 software (Carl Zeiss) were used for noise reduction, projection, and analysis of the fluorescence images. Fluorescence signals in the figures and supplementary movies were pseudo-colored. Supplementary Table 2 summarizes the color codes for different markers in the illustrations.

### Airyscan super-resolution imaging of human pancreas
Prior to Airyscan, standard confocal imaging with x10 (C-Apochromat ×10/0.45 W, Carl Zeiss), ×25 (LD LCI Plan-Apochromat ×25/0.8 Imm Corr DIC, Carl Zeiss), and/or ×40 (LD C-Apochromat ×40/1.1 W, Carl Zeiss) objectives were performed to evaluate the lobular environment and to identify the area of interest (PanIN in Fig. 5 and ganglion in Fig. 7). In Airyscan, super-resolution imaging with ×40 (Plan-Apochromat ×40/1.3 Oil DIC) and ×63 (Plan-Apochromat ×63/1.4 Oil DIC) objectives were sequentially applied (laser power, 3–5%; master gain, 650–780 V; pixel dwell time, ×40 objective: 2.06 μs, ×63 objective: 3.82 μs; additional parameters are listed in Supplementary Fig. 7). The first round (×40) was to integrate the super-resolution images with the panoramic view of pancreas; the second round (×63) offers detailed examination of the subcellular structures. The multiple rounds of fluorescence imaging were made possible by the antifade feature of A-ha-based fluorescence imaging (Fig. 6a–c).

### Antifade test of fluorescence microscopy
Human pancreas vasculature (500-μm specimen) labeled with Alexa Fluor (AF) −647-conjugated anti-CD31 primary antibody (direct immunohistochemistry, sensitive to photobleaching) was used in Zeiss LSM 800 time-series imaging to evaluate the decrease in fluorescence (*Histogram* function of Zen software, Carl Zeiss) against cycles of excitation and emission (frames taken). In parallel, the less sensitive DAPI labeling of nuclei (chemical stain) and indirect immunohistochemistry (anti-S100B labeled with multiple AF-546-secondary antibodies) were used as the labeling controls to monitor the fluorescence signals and imaging process (×40 objective, LD C-Apochromat ×40/1.1 W, Carl Zeiss; ~30 μm under tissue surface; 500 cycles; 90 min). The triple-labeled specimens were embedded in A-ha or immersed in the clearing reagent Ce3D (aqueous solution, liquid control; BioLegend, San Diego, CA, USA; 427702) or CytoVista (organic solvent, liquid control; ThermoFisher, V11300) to perform the antifade test.

### Multimodal A-ha/3D and clinical/2D pancreas histology
Vibratome sections of human pancreas (350 μm in thickness; Fig. 5a, b) were examined with stereomicroscopy to detect the potential duct lesion[41]. Afterward, the area of interest was examined by clinical 2D histology (H&E stain of microtome section, gold standard) to confirm the PanIN lesion (Fig. 5d). Once confirmed, fluorescence labeling of the adjacent vibratome sections (Fig. 5e–j, k–o) was performed to reveal the neurovascular networks and duct-islet cell clusters in the same microenvironment. Both the standard confocal and Airyscan microscopy (super-resolution mode, Carl Zeiss) were used to characterize the peri-lesional microenvironment.

### Statistics and reproducibility
The quantitative values are presented as means with standard deviation (except Fig. 1a). Statistical differences were determined by the two-sided unpaired Student's $t$ test. Differences between groups were considered statistically significant when $p < 0.05$. No statistical method was used to predetermine sample size. No data were excluded from the analyses. The experiments were not randomized. The investigators were not blinded to allocation during experiments and outcome assessment.

### Reporting summary
Further information on research design is available in the Nature Portfolio Reporting Summary linked to this article.

## Data availability
Considering the large size of 3D confocal and Airyscan super-resolution image files, the raw datasets are available from the corresponding author S.C.T. on request. Requests will be answered within 3 weeks to arrange file transfer. Source data are provided with this paper.

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

## Acknowledgements

This work was supported in part by grants from Taiwan Ministry of Science and Technology (MOST 111-2740-B-001-005) to C.N.S. and S.C.T., MOST 111-2314-B-007-005-MY2 to S.C.T., Taiwan Academia Sinica (AS-107-TP-L15) to C.N.S., Y.W.T., and S.C.T., Taiwan National Health Research Institutes (NHRI-EX111-10922EI and NHRI-EX112-11225EI) to Y.W.T and S.C.T., and National Tsing Hua University, Taiwan (intramural 110F7MAKE1) to S.C.T. The authors are grateful for the support from the confocal imaging core in National Tsing Hua University, Taiwan, which is sponsored by the Ministry of Science and Technology (MOST 110-2731-M-007-001). We thank Dr. S.-Ja Tseng at the Graduate Institute of Oncology, National Taiwan University and Dr. Ho-Hsiu Chou at the Department of Chemical Engineering, National Tsing Hua University, Taiwan, for FT-IR analysis of A-ha copolymer.

## Author contributions

All authors contributed to the study concept and design; F.T.H. and H.J.C. contributed to A-ha copolymer synthesis and characterization; Y.W.T., H.P.C., C.Y.L., C.C.C., and Y.M.J. contributed to human pancreas acquisition and preparation for 3D imaging; F.T.H., H.J.C., T.H.H., and S.J.P., contributed to mouse tissue preparation; F.T.H., H.J.C., Y.H.C., S.J.P., M.H.C., T.H.H., L.W. L., C.N.S., and S.C.T. contributed to 3D tissue imaging and image presentation; and S.C.T. contributed to drafting of the manuscript. C.N.S., Y.W.T., and S.C.T. obtained funding. All authors contributed to data analysis and interpretation of data, revised the manuscript critically for intellectual content, and approved the final version of the manuscript. S.C.T. is the guarantor of this work and, as such, had full access to all the data in the study and takes responsibility for the integrity of the data and the accuracy of the data analysis.

## Competing interests

The authors declare no competing interests.
