## [Peer Review File · Nature Communications]

Transparent tissue in solid state for solvent-free and antifade 3D imagingREVIEWER COMMENTS

Reviewer #1 (Remarks to the Author):

Mei-Hsin Chung and his colleague developed a novel tissue-clearing method that allows simultaneous refractive index adjustment and embedding. By using this method, they have achieved higher photochemical stability of the fluorescent dye compared to Ce3D and CytoVista. They applied this method to mouse kidneys and the human pancreas to make detailed observations.

Major concerns:

Line 72-75: Fig4 shows that the photochemical stability of AF647 increased compared with Ce3D and CytoVista, but the photochemical stability of DAPI decreased compared with Ce3D. Increasing the photochemical stability of A-ha copolymer seems to depend on dyes. Authors should also measure the photochemical stability of other dyes in this solid and it would be better to compare it with more commonly used tissue-clearing methods (iDisco, CLARITY, and CUBIC).

Line 91-94: They claim that polymerization at room temperature reduces photobleaching, but it is unclear whether there is no data showing the relationship between polymerization and photobleaching.

Minor concerns:

Fig 1.d-j, Fig 2 b-e: There is no scale bar.

Supplementary Fig. 3 a: Please describe how you measured the size. You must specify whether distance, area, or volume.

Reviewer #2 (Remarks to the Author):

In this study, the authors established a new optical tissue clearing method in solid state. Because present optical clearing methods are liquid-based methods, the solid-based methods are quite unique and give some advantages for the application to 3D bioimaging. Besides, this manuscript contains many beautiful fluorescence images. Thus, this is worth to be published in Nature Commun. However, there are several points to be improved, as follows.

1) Applications of some copolymer hydrogels to optical tissue clearing were reported (Ono et al, J. Biomed. Mater. Res. B, 2019; Kojima et al, Macromol. Biosci. 2021 etc.). Cite these papers and explain them in the Introduction.

2) Poly(acrylamide-co-n-hydroxymethyl acrylamide) at the 1/1 mole ratio was used in this study. Fig 1 shows that refractive indices of polyacrylamide and poly(n-hydroxymethyl acrylamide) are similar refractive index, suggesting that high concentration of polyacrylamide possibly shows the similar effects in the optical tissue clearing. The authors should do the same experiments using polyacrylamide (homopolymer) and copolymers with different mole ratios to clarify the effect of n-hydroxymethyl acrylamide. And, discussion of roles of n-hydroxymethyl acrylamide should be added. This is the most important point.

3) Some figure captions, especially Fig. 4, are difficult to understand. For example, each color in fluorescence images should be shown in all figures and panels. In Fig.3, glucagon and CK7 are correlated to vesicle and cytokeratin, respectively. The explanation is required for broad readers.

4) Conclusion should be added at the end of the discussion part.

Reviewer #3 (Remarks to the Author):

This study presents a novel method using a solvent-free approach to embed tissues and immunodetect the different cell types. Authors show that their new method of embedding, using a high-n acrylamide-based copolymer, protects from photobleaching as well. This method is of high

interest and seems simple to apply. I only have a couple of recommendations to improve the manuscript. Particularly some important aspects of the protocol are not described and some quantitative assessment of the intracellular expansion caused by the polymer. More specifically:

1) I could not find the laser power, master gain, pixel dwell and type of scanning performed when obtaining the Airyscan images. Such information is needed to determine the extend of protection from photobleaching.

2) There are images of vesicles inside endocrine cells of the pancreas. Labeling other organelles and measuring their size would help to quantify how much the polymer expanded the tissue. In addition, it would be nice to determine whether the expansion allowed to separate subcellular compartments sufficiently to discern the different localization of proteins that cannot be resolved with confocal microscopy (i.e. outer membrane versus inner membrane proteins). These should be compared side by side with other similar approaches not using this new polymer.

3) A video showing a 3D reconstruction, stacking the images collected across a Z stack, would be nice.

4) Chemical safety issues about shipping these tissues should be discussed, to facilitate the work to researchers that want to reproduce this method to ship samples to other labs.

Reviewer #4 (Remarks to the Author):

The manuscript „Transparent tissue in solid state for 3D imaging: solvent-free and antifade“ of Hsiao and coauthors presents an interesting novel embedding technique for histopathology and tissue biology using high-n-acrylamide-based copolymers already known for their use in polyacrylamide gels in standard protein electrophoresis. Aim of this study was to develop a transparent, liquid-free tissue condition enabling 3D-imaging in optimal resolution for different applications such as histochemistry and fluorescence imaging.

The manuscript is well written and includes several high quality figures and numerous videos. The technique is simple, but novel, and well described. As acrylamide is already widely used, the novel technique may be of significant use in 3 D microscopy and may be easily reproducible.

Concerning the method itself, the quality of tissue integrity and preservation is overall very good: the study used different mouse and human tissues including brain, kidney, liver and pancreas, the latter representing tissues with especially high risk of autolysis due to endogenous enzymes such as proteases. The reviewer sees no flaws in data analysis, interpretation and conclusions.

As minor point, the authors should further discuss durability of tissue in polyacrylamide gels and add a statement on antibody penetration in thicker tissue slides as used in their 3D models. In addition, the authors should state on super resolution in their model, e.g. on imaging of cell organelles (nuclear pore complexes, mitochondria etc).

RESPONSE TO REVIEWERS' COMMENTS

Reviewer #1 (Remarks to the Author)

Comment:

Mei-Hsin Chung and his colleague developed a novel tissue-clearing method that allows simultaneous refractive index adjustment and embedding. By using this method, they have achieved higher photochemical stability of the fluorescent dye compared to Ce3D and CytoVista. They applied this method to mouse kidneys and the human pancreas to make detailed observations.

Major concerns:

Line 72-75: Fig4 shows that the photochemical stability of AF647 increased compared with Ce3D and CytoVista, but the photochemical stability of DAPI decreased compared with Ce3D. Increasing the photochemical stability of A-ha copolymer seems to depend on dyes. Authors should also measure the photochemical stability of other dyes in this solid and it would be better to compare it with more commonly used tissue-clearing methods (iDisco, CLARITY, and CUBIC).

Response

We thank Reviewer 1 for the comment. We focused on the Alexa Fluor (AF) dyes because they have been widely applied in indirect and direct immunohistochemistry for 2D tissue and cellular analyses. Here, for 3D imaging, our goal is to use the solid A-ha copolymer to maintain the stability of AF dyes; thus, their fluorescence signals can be consistently detected in the in-depth Airyscan super-resolution imaging without the false negative interference due to photobleaching. The result is presented in **Fig. 4l-p** (>11 hours for two rounds of Airyscan, **Supplementary Fig. 7**) and **Supplementary Movie 10-12**.

In the new **Supplementary Fig. 5a-d**, we applied direct immunohistochemistry (CD31-conjugated-AF-647) and the 500-frame antifade test to compare the dye stability in the A-ha copolymer vs. the high-*n* liquids used in iDISCO (dibenzyl ether), CLARITY (FocusClear), and CUBIC (sucrose/urea/triethanolamine solution). The 500-frame antifade test was performed on the 4- μ m microtome section of human pancreas; thus, only the chemical environment, not tissue clearing efficiency, affects the fluorescence detection. On average, the fluorescence signals decreased by $8\pm 6\%$ in the A-ha copolymer and $47\pm 7\%$, $75\pm 5\%$, and $53\pm 3\%$ in the solutions of dibenzyl ether, FocusClear, and sucrose/urea/triethanolamine, respectively ($p < 0.001$ vs. A-ha; six repeats). **Supplementary Movie 9** shows the side-by-side and frame-by-frame comparison of the four conditions. This information is added on page 11, first paragraph (redlined) and in the legend of **Supplementary Fig. 5**.

Also, in **Supplementary Fig. 5e, f**, we show that, after 500 frames were taken from the DAPI, Hoechst, and propidium iodide-labeled nuclei of human pancreas, the decreases in nuclear signals in the A-ha copolymer and Ce3D solution are all less than 20%. The result indicates that the three nuclear dyes are intrinsically stable in fluorescence imaging.

Because DAPI and propidium iodide are intercalating agents and their quantum yield drastically increase after binding to DNA (enhanced by one order of magnitude), we do not know if their photobleaching is primarily caused by the diffusional contacts between the dyes and reactive species (e.g., oxygen), as that of the indicator dyes used in antibody staining, or by the change in the dye-DNA complex.

In this study, because the decrease in nuclear signals does not affect 3D/Airyscan super-resolution

imaging, we did not investigate the photobleaching of nuclear dyes. Rather, we used DAPI as a positive control to monitor the fluorescence signal detection in the 500-frame antifade test (**Fig. 4a-i** and **Supplementary Fig. 5a-d**).

Comment:

Line 91-94: They claim that polymerization at room temperature reduces photobleaching, but it is unclear whether there is no data showing the relationship between polymerization and photobleaching.

Response

We thank Reviewer 1 for the comment. To avoid confusion, on page 6, second paragraph, we rewrote the advantage of using photo-polymerization to synthesize polyacrylamide for tissue embedding, clearing, and 3D imaging (redlined). The following is the revised writing (page 6, second paragraph, line 113-115).

“The high-*n* acrylamide-based polymers, in particular, provide a unique opportunity for simultaneous tissue embedding and clearing at a solid state to replace the high-*n* liquid in 3D imaging. Importantly, because the synthesis of polyacrylamide can be controlled by ultraviolet radiation, the photo-polymerization can be directly performed on the microscope glass to integrate the processes of tissue embedding, clearing, and 3D imaging with the same polymerized tissue slide. Here, using an efficient two-step approach, we establish a new class of tissue clearing method via tissue embedding in the high-*n* copolymer of acrylamide and n-hydroxymethyl acrylamide (or A-ha copolymer) for 3D fluorescence imaging with minimal photobleaching.”

Comment:

Minor concerns:

Fig 1.d-j, Fig 2 b-e: There is no scale bar.

Response

We thank Reviewer 1 for the comment. Scale bars have been added to the images in **Fig. 1d-j** and **Fig. 2b-e**.

Comment:

Supplementary Fig. 3a: Please describe how you measured the size. You must specify whether distance, area, or volume.

Response

We thank Reviewer 1 for the comment. The kidney specimen was first immersed in PBS (between coverslips with a 500- μ m spacer) and then imaged via stereomicroscopy (Carl Zeiss, SteREO Discovery.V12) to record the pixels occupied by the tissue (defined as 100). Afterward, the tissue was cleared in the A-ha copolymer (or clearing liquid) and then imaged again under the same microscope. The number of pixels recorded in the clearing condition was divided by that in PBS to estimate the relative tissue size after optical clearing (y axis of **Supplementary Fig. 3a**). In the process, the same background “Mouse kidney” was placed under the specimen to serve as the size standard (relative to tissue) and to demonstrate the change in tissue transparency. This description has been added to the legend.

Scale bars have also been added to the images in **Supplementary Fig. 3b-d**.

Reviewer #2 (Remarks to the Author)

Comment:

In this study, the authors established a new optical tissue clearing method in solid state. Because present optical clearing methods are liquid-based methods, the solid-based methods are quite unique and give some advantages for the application to 3D bioimaging. Besides, this manuscript contains many beautiful fluorescence images. Thus, this is worth to be published in Nature Commun. However, there are several points to be improved, as follows.

1) Applications of some copolymer hydrogels to optical tissue clearing were reported (Ono et al, J. Biomed. Mater. Res. B, 2019; Kojima et al, Macromol. Biosci. 2021 etc.). Cite these papers and explain them in the Introduction.

Response

We thank Reviewer 2 for the positive assessment of our work and the suggestion. The two papers have been cited in the revised introduction. On page 6, second paragraph, we wrote:

“Among the tissue-friendly polymers, the low-density polyacrylamide gel has long been used in electrophoresis and recently in the CLARITY/hydrogel approach to support the lipid-extracted tissue in the clearing solution for 3D microscopy²⁰⁻²². By adjusting the hydrogel’s composition, the chemical environment of the tissue-hydrogel complex, such as the electrostatic interactions, can be engineered to create a favorable condition for tissue clearing^{23,24}.”

Comment:

2) Poly(acrylamide-co-n-hydroxymethyl acrylamide) at the 1/1 mole ratio was used in this study. Fig 1 shows that refractive indices of polyacrylamide and poly(n-hydroxymethyl acrylamide) are similar refractive index, suggesting that high concentration of polyacrylamide possibly shows the similar effects in the optical tissue clearing. The authors should do the same experiments using polyacrylamide (homopolymer) and copolymers with different mole ratios to clarify the effect of n-hydroxymethyl acrylamide. And, discussion of roles of n-hydroxymethyl acrylamide should be added. This is the most important point.

Response

We thank Reviewer 2 for this comment. There were two considerations that led to the use of poly(acrylamide-co-n-hydroxymethyl acrylamide) at the ultrahigh monomer-to-water ratio (6.5 to 1, or 86.7% monomer mass fraction, wt/wt; water at 13.3%) in this research.

1. Maximum concentration of acrylamide in water and gel-like property of polyacrylamide

To form a high- n polymer, our approach is to increase the concentration of monomer (i.e., solute) in water, which leads to the formation of high- n polymer after UV radiation (**Fig. 1a** and Gladstone-Dale equation: $(n-1)/\rho = \text{constant}$; higher ρ , and thus higher n).

In the binary acrylamide-water system, the limitation of n is set by the maximum concentration of acrylamide in water (i.e., solubility), which is at ~60% wt/wt at room temperature. Unfortunately, at this monomer concentration, the homopolymer polyacrylamide is gel-like, which develops cracks in air (loss of water) and becomes opaque (**Fig. A-C** below). This situation is similar to that in the images shown in **Supplementary Fig. 1b** and **b'**, in which 75% wt/wt of A-ha monomers were used, leading to the formation of gel-like A-ha copolymer.

Polyacrylamide (homopolymer, control)

Fig. A-C. Opaque polyacrylamide homopolymer in air (photo-polymerized at 40%, 50%, and 60% wt/wt). Acrylamide reaches solubility limit at 60% wt/wt in water at room temperature.

In the CLARITY/hydrogel condition, the tissue-polymer complex is not solvent-free. The complex requires a high- n liquid to avoid opacity caused by dehydration. The two papers listed in the reviewer's first comment are in this condition.

2. Ternary system of A-ha monomer solution and entropy of mixing

To break the limit of monomer's solubility in water, we apply the concept of "entropy of mixing," which states: $\Delta S_{\text{mix}} = -nR (x_A \ln x_A + x_B \ln x_B)$ for a binary system (S: entropy; n : total number of moles; R : constant; x_i : mole fraction of component; ideal situation) [cited from: Introduction to Chemical Engineering Thermodynamics; Smith, Van Ness, Abbott, and Swihart, 8th Edition, McGraw-Hill, 2017].

In polymer thermodynamics, the most common way to increase ΔS_{mix} (i.e., solubility of monomer) is to increase temperature (i.e., randomness). However, this deviates from our goal of polymerization at room temperature.

Alternatively, to increase the randomness, we added a second monomer, n-hydroxymethyl acrylamide, to the solution, which contributes to the entropy $\Delta S_{\text{mix}} = -nR (x_{\text{H}_2\text{O}} \ln x_{\text{H}_2\text{O}} + x_a \ln x_a + x_{\text{ha}} \ln x_{\text{ha}})$. Adding ha dilutes acrylamide but increases the overall monomer mass fraction $x_a + x_{\text{ha}}$ in solution, which strengthens the polymer structure after photo-polymerization.

The result is demonstrated in **Supplementary Fig. 1a**, in which the blue area (Region II) forms rigid and transparent A-ha copolymer (**Supplementary Fig. 1c**) vs. the gel-like structure formed at 75% wt/wt of A-ha monomers in **Supplementary Fig. 1b, b'**.

In the revised manuscript, we added **Fig. A-C** to **Supplementary Fig. 1** and indicate that the homopolymer polyacrylamide (photo-polymerized at 40%, 50%, and 60% wt/wt) becomes opaque in air.

We decided not to include the aforementioned thermodynamics concept because the information may

distract readers without a physical chemistry background. The information, however, can be added as a note in **Supplementary Fig. 1** if Reviewer 2 thinks it is necessary.

Comment:

3) Some figure captions, especially Fig. 4, are difficult to understand. For example, each color in fluorescence images should be shown in all figures and panels. In Fig.3, glucagon and CK7 are correlated to vesicle and cytokeratin, respectively. The explanation is required for broad readers.

Response

We thank Reviewer 2 for the suggestion. We have modified the legends of **Fig. 3** and **4**.

In **Fig. 3**, the legend of panel *j-n* has been revised as:

“(j-n) Peri-lesional endocrine pancreas remodeling preserved and revealed in A-ha. Panel j (vibratome section adjacent to c) shows the duct lesion. The lesion and the duct-islet cell cluster are enlarged in k and l (arrows). White, DAPI, nuclei; blue, insulin, β-cells; magenta, glucagon, α-cells; green, cytokeratin 7 (CK7), duct cells. Cytokeratin filaments and vesicles are revealed by Airyscan super-resolution imaging with 40x objective (l). The asterisk in k is enlarged in m and further magnified in n (63x objective) to confirm the glucagon[±] vesicles of α-cell (magenta) and cytokeratin filaments of duct cell (CK7[±], green). The panels from *a* to *n* (same lesion environment) illustrate the multimodal, multidimensional, and multiscale approaches of human pancreas imaging with A-ha copolymer.”

In **Fig. 4**, the legend of panel *j-p* has been revised as:

“(j-p) Airyscan 3D super-resolution imaging of human pancreas. Panel j (vibratome section) and k (confocal image) are derived from the same lobule. Arrows indicate that a ganglion (S100B[±], green) around a vascular pathway (CD31[±], red) is preserved in the A-ha copolymer. White, DAPI, nuclei. The ganglion is enlarged in l-p via in-depth Airyscan to illustrate the glial-neuronal association. l-n are 2D image and 3D projection of the ganglion (40x objective). Airyscan with 63x objective (o, p) detects and confirms the dimly stained neuronal nucleus (nucleolus at the center). Cyan arrows in k, m, p indicate the lobule-to-nucleolus magnification of human pancreas. Also see **Supplementary Movie 10-12** for in-depth Airyscan of the glial-neuronal association.”

Comment:

4) Conclusion should be added at the end of the discussion part.

Response

We thank Reviewer 2 for this comment. We have added the conclusion at the end of the discussion section.

On page 13, second paragraph, we wrote:

“In conclusion, before this research, 3D tissue imaging has been limited by solvent and dye instability in a liquid environment. Using the Gladstone-Dale equation as a design concept, we develop a solid (solvent-free) high-*n* A-ha copolymer to embed mouse and human tissues for clearing and antifade imaging. The new microscopic environment offers a friendly and versatile platform that integrates with standard fluorescent dye labeling, clinical 2D histology, and 3D super-resolution imaging for investigation of the cellular structures and neurovascular networks in health and disease.”

Reviewer #3 (Remarks to the Author)

Comment:

This study presents a novel method using a solvent-free approach to embed tissues and immunodetect the different cell types. Authors show that their new method of embedding, using a high-n acrylamide-based copolymer, protects from photobleaching as well. This method is of high interest and seems simple to apply. I only have a couple of recommendations to improve the manuscript. Particularly some important aspects of the protocol are not described and some quantitative assessment of the intracellular expansion caused by the polymer. More specifically:

1) I could not find the laser power, master gain, pixel dwell and type of scanning performed when obtaining the Airyscan images. Such information is needed to determine the extend of protection from photobleaching.

Response

We thank Reviewer 3 for the positive assessment of our work and the suggestion. The information about the laser power (3-5%), master gain (650-780V), pixel dwell time (40x objective: 2.06 μ s; 63x objective: 3.82 μ s), and additional acquisition parameters of Airyscan is provided in the **Methods** section (page 16, second paragraph) and listed in **Supplementary Fig. 7**.

Comment:

2) There are images of vesicles inside endocrine cells of the pancreas. Labeling other organelles and measuring their size would help to quantify how much the polymer expanded the tissue. In addition, it would be nice to determine whether the expansion allowed to separate subcellular compartments sufficiently to discern the different localization of proteins that cannot be resolved with confocal microscopy (i.e. outer membrane versus inner membrane proteins). These should be compared side by side with other similar approaches not using this new polymer.

Response

We thank Reviewer 3 for the suggestion. The mouse and human tissues that we tested in this study were formaldehyde-fixed. It has been widely reported that the fixation process leads to tissue shrinkage (e.g., ~30% for human colorectal specimens [1]). Unlike the organic clearing solvent, which tends to aggravate the shrinkage [2-4], the hydrophilic A-ha copolymer mildly expands the tissue in embedding (18 \pm 2%; **Supplementary Fig. 3a**). Inside the tissue, we expect that the cellular components (including organelles) and extracellular matrix also expand at a similar range, leading to the overall tissue expansion.

Technically, because 3D Airyscan is a new approach to investigate cellular structures in thick human specimens, we are very careful in choosing targets to avoid misrepresentation of this work. Based on our prior experience, multiple factors, including 3D immunolabeling, refractive index matching, and point spread function, are involved in the 3D signal acquisition and can affect the observed fluorescence. We chose to examine the human pancreas because: i) the organ is important in diabetes and pancreatic cancer research and ii) our knowledge about the pancreas microstructure, vasculature, and innervation helps avoid the false negative and false positive results that we previously observed in 3D pancreas imaging (e.g., false positive signals from residual blood in tissue [5]).

Organelle (e.g., mitochondria of β -cell) imaging in health and disease (e.g., diabetes) are certainly important topics in future applications of 3D super-resolution imaging. We appreciate the reviewer's suggestion. But in the meantime, we believe this new imaging approach will create greater impact by informing experts of different fields to study their preferred targets. This will allow more high-

quality images of human tissues and cells to become available for comparison and analysis.

References:

- [1] Goldstein, N.S., Soman, A. & Sacksner, J. Disparate surgical margin lengths of colorectal resection specimens between in vivo and in vitro measurements. The effects of surgical resection and formalin fixation on organ shrinkage. *Am J Clin Pathol* 111, 349-351 (1999).
- [2] Erturk, A. et al. Three-dimensional imaging of solvent-cleared organs using 3DISCO. *Nat Protoc* 7, 1983-1995 (2012).
- [3] Pan, C. et al. Shrinkage-mediated imaging of entire organs and organisms using uDISCO. *Nat Methods* 13, 859-867 (2016).
- [4] Bekkouche, B.M.B., Fritz, H.K.M., Rigosi, E. & O'Carroll, D.C. Comparison of transparency and shrinkage during clearing of insect brains using media with tunable refractive index. *Front Neuroanat* 14, 599282 (2020).
- [5] Chien, H.J. et al. Human pancreatic afferent and efferent nerves: mapping and 3-D illustration of exocrine, endocrine, and adipose innervation. *Am J Physiol Gastrointest Liver Physiol* 317, G694-G706 (2019).

Comment:

3) A video showing a 3D reconstruction, stacking the images collected across a Z stack, would be nice.

Response

We thank Reviewer 3 for the suggestion. We added a new **Supplementary Movie 11** entitled “**3D Airyscan and image reconstruction of human intrapancreatic ganglion**” to the revised manuscript.

The first part of the movie (00:00-01:08) shows in-depth Airyscan of intrapancreatic ganglion. Four neurons with dimly stained nuclei are revealed (00:22-00:42). The second part of the movie (01:09-01:52) shows 360-degree projection of the S100B-labeled ganglion.

The pancreas was labeled with DAPI (white), anti-S100B (green, standard IHC), and anti-CD31 (red, AF-647-conjugated primary antibody). Dimensions of the image stack: 160 (x) × 160 (y) × 130 (z, depth) μm.

Comment:

4) Chemical safety issues about shipping these tissues should be discussed, to facilitate the work to researchers that want to reproduce this method to ship samples to other labs.

Response

We thank Reviewer 3 for the suggestion. In the revised Discussion section, a new paragraph has been added to discuss the chemical safety and tissue durability in the A-ha copolymer. On page 13, first paragraph, we wrote:

“Regarding safety in polymer synthesis, we would like to stress that the monomers of A-ha copolymer -- acrylamide and n-hydroxymethyl acrylamide -- are toxins when in solution (especially acrylamide), so care must be taken to avoid direct contact with the monomer solution. Once polymerized, similar to the polyacrylamide gel used in electrophoresis, the A-ha copolymer is chemically stable (long shelf life) with strong mechanical strength (provided by ultrahigh monomer content, Fig. 1a) to protect the cleared 3D specimen in transport and imaging.”

Reviewer #4 (Remarks to the Author)

Comment:

The manuscript “Transparent tissue in solid state for 3D imaging: solvent-free and antifade“ of Hsiao and coauthors presents an interesting novel embedding technique for histopathology and tissue biology using high-n-acrylamide-based copolymers already known for their use in polyacrylamide gels in standard protein electrophoresis. Aim of this study was to develop a transparent, liquid-free tissue condition enabling 3D-imaging in optimal resolution for different applications such as histochemistry and fluorescence imaging.

The manuscript is well written and includes several high quality figures and numerous videos. The technique is simple, but novel, and well described. As acrylamide is already widely used, the novel technique may be of significant use in 3 D microscopy and may be easily reproducible.

Response

We thank Reviewer 4 for the positive assessment of our work and highlighting the significance of this research.

Comment:

Concerning the method itself, the quality of tissue integrity and preservation is overall very good: the study used different mouse and human tissues including brain, kidney, liver and pancreas, the latter representing tissues with especially high risk of autolysis due to endogenous enzymes such as proteases. The reviewer sees no flaws in data analysis, interpretation and conclusions.

Response

We thank Reviewer 4 for stating the quality of this research.

Comment:

As minor point, the authors should further discuss durability of tissue in polyacrylamide gels and add a statement on antibody penetration in thicker tissue slides as used in their 3D models. In addition, the authors should state on super resolution in their model, e.g. on imaging of cell organelles (nuclear pore complexes, mitochondria etc).

Response

We thank Reviewer 4 for the suggestions.

On page 13, first paragraph, line 280-282, we added a description about the durability of tissue in A-ha copolymer: “Once polymerized, similar to the polyacrylamide gel used in electrophoresis, the A-ha copolymer is chemically stable (long shelf life) with strong mechanical strength (provided by ultrahigh monomer content, Fig. 1a) to protect the cleared 3D specimen in transport and imaging.”

On page 12, third paragraph, we added the discussion about using vascular (AF-647-anti-CD31) and nuclear (DAPI) labeling as positive controls to monitor antibody and dye penetration. In addition, this approach also provides the imaging targets to compare different super-resolution methods. We wrote:

“In 3D signal detection, because false-negative results are frequently encountered in fluorescent immunohistochemistry, the antifade vascular (anti-CD31) and nuclear (DAPI) labeling developed in this research provides the important positive controls for clinical tissue examination -- due to their abundance -- to monitor the penetration of antibody and indicator dye in 3D labeling. In the imaging facility core, the antifade controls also offer a suitable chemical environment to compare different super-resolution methods (e.g., Airyscan vs. stimulated emission depletion, or STED, microscopy) via a standard 3D specimen (e.g., Supplementary Fig. 6c).”

We did not comment on 3D imaging of cell organelles because we lack image data to provide insights into this topic. However, to confirm the A-ha-based antifade labeling, we performed a feasibility test with Leica 3D STED microscopy of human pancreas (off-site imaging via shipment of slide shown in **Supplementary Fig. 6c**). We thus feel comfortable commenting on this topic at the end of the paragraph.

Again, we thank the editor and reviewers for the time spent reviewing our work and the important comments and suggestions that improve the quality of this manuscript.

REVIEWERS' COMMENTS

Reviewer #1 (Remarks to the Author):

Mei-Hsin Chung and his colleague developed a new tissue-clearing method using a polymerizing agent called A-ha copolymer. Because this method is a solid-based method, it has some advantages such as being resistant to fading.

All points raised in the first version of the manuscript have been properly addressed by the authors, providing additional data.

Reviewer #2 (Remarks to the Author):

The authors replied to my comments properly.

Reviewer #3 (Remarks to the Author):

The authors successfully addressed the major concerns.

Reviewer #4 (Remarks to the Author):

The critic points have been fully addressed and the manuscript may be accepted for publication

RESPONSE TO REVIEWERS' COMMENTS

Reviewer #1 (Remarks to the Author)

Comment:

Mei-Hsin Chung and his colleague developed a new tissue-clearing method using a polymerizing agent called A-ha copolymer. Because this method is a solid-based method, it has some advantages such as being resistant to fading.

All points raised in the first version of the manuscript have been properly addressed by the authors, providing additional data.

Response

We thank Reviewer 1 for the positive assessment of our manuscript revision.

Reviewer #2 (Remarks to the Author)

Comment:

The authors replied to my comments properly.

Response

We thank Reviewer 2 for the positive assessment of our manuscript revision.

Reviewer #3 (Remarks to the Author)

Comment:

The authors successfully addressed the major concerns

Response

We thank Reviewer 3 for the positive assessment of our manuscript revision.

Reviewer #4 (Remarks to the Author)

Comment:

The critic points have been fully addressed and the manuscript may be accepted for publication

Response

We thank Reviewer 4 for the positive assessment of our manuscript revision.

We thank the editor and reviewers for the time spent reviewing our work. The important comments and suggestions have greatly improved the quality and presentation of this manuscript.